# A scoping review of recommendations in the English language on conducting research with trauma-exposed populations since publication of the Belmont report; thematic review of existing recommendations on research with trauma-exposed populations

Kevin Jefferson[1]* , Kaitlyn K. Stanhope[2] , Carla Jones-Harrell[3‡], Aimée Vester[3‡], Emma Tyano[4‡], Casey D. Xavier Hall[5,6‡]

1 Independent Researcher, Atlanta, Georgia, United States of America, 2 Emory University School of Medicine, Atlanta, Georgia, United States of America, 3 Rollins School of Public Health, Emory University, Atlanta, Georgia, United States of America, 4 Nell Hodgson Woodruff School of Nursing, Emory University, Atlanta, Georgia, United States of America, 5 Institute for Sexual and Gender Minority Health and Well-being, Northwestern University, Evanston, Illinois, United States of America, 6 Medical Social Sciences, Feinberg School of Medicine, Northwestern University Chicago, Chicago, Illinois, United States of America

☯ These authors contributed equally to this work.
‡ These authors also contributed equally to this work.
* kevinjres@gmail.com

## Abstract

### Objective

To identify recommendations for conducting public health research with trauma-exposed populations.

### Methods

Researchers searched Embase, PubMed, Scopus, Web of Science, Open Grey, and Google Scholar for recommendations. Trauma that causes psychological impact was our exposure of interest and we excluded clinical articles on treating physical trauma. We reviewed titles and abstracts of 8,070 articles and full text of 300 articles. We analyzed recommendations with thematic analysis, generated questions from the existing pool of recommendations, and then summarized select gaps.

### Results

We abstracted recommendations from 145 articles in five categories: community benefit, participant benefit, safety, researcher well-being, and recommendations for conduct of trauma research.

**Data Availability Statement:** All relevant data are within the manuscript and its Supporting information files.

**Funding:** KS, No grant number, Laney Graduate School Professional Development Support Funds, https://www.gs.emory.edu/professional-development/pds/index.html KJ's affiliation is not a commercial affiliation and he does not receive funds in connection with it. Laney Graduate School Professional Development Support Funds providing funding for Covidence software to KS but did not have any additional role in the study design, data collection and analysis, decision to publish, or preparation of the manuscript. The specific roles of these authors are articulated in the 'author contributions' section.

**Competing interests:** The authors have no competing interests to declare. KJ's stated affiliation is not a commercial affiliation. There are no declarations relating to employment, consultancy, patents, products in development, or marketed products, etc. This does not alter our adherence to PLOS ONE policies on sharing data and materials.

## Conclusions

Gold standards to guide the conduct of trauma-informed public health research do not yet exist. The literature suggests participation in trauma research is not inherently harmful, and current recommendations concern using research to benefit communities and participants, protecting participants and researchers from harm, and improving professional practice. As public health researchers increasingly analyze trauma as a determinant of health, gold standards for the conduct of trauma-informed public health research would be appropriate and timely.

## Introduction

Increasingly, public health scholars research trauma as a determinant of health. Moreover, in addition to a growing body of work studying the effect of trauma on physiological and mental health outcomes [1–3], large national surveys (e.g., the Behavioral Risk Factor Surveillance System) [4], ongoing surveillance systems [5], and primary care practices [6–8] regularly assess trauma exposures, bringing increased attention to its prevalence in the United States. According to the Substance Abuse and Mental Health Services Administration (SAMHSA), "individual trauma results from an event, series of events, or set of circumstances that is experienced by an individual as physically or emotionally harmful or life threatening and that has lasting adverse effects on the individual's functioning and mental, physical, social, emotional, or spiritual well-being." [9,pg. 7]. Trauma is common in the United States, with as much as 80% of the population reporting at least one lifetime trauma experience of war, maltreatment in childhood, assault, injury or shocking event, or unexpected death [10]. With the understanding that trauma is highly prevalent, the study of trauma and research involving trauma-exposed populations is not restricted to any one content area. Rather, public health researchers across a broad range of content need guidance for appropriate trauma-informed practices.

In the fields of medicine, social work, education, and other services, organizations have embraced principles of trauma-informed care to meet the needs of trauma-exposed individuals [11–14]. According to SAMHSA, trauma-informed care approaches should: realize the impact of trauma, recognize and respond to signs and symptoms of trauma in all aspects of practice, and avoid retraumatization [9]. This guidance informs interactions with trauma-exposed individuals, but within public health it may be beneficial to think of trauma using a social ecological approach [15] attending to oppression-based trauma [16], where not only individuals, but entire communities, may be impacted by systemic, cultural, historical, or racial trauma [17–21]. It is not clear how existing trauma-informed practice approaches apply to research, whether public health researchers can adopt them as is or need research-specific guidelines, or how to address systemic, cultural, historical, or racial trauma in public health research.

Currently, the most widely used guiding principles for the design and implementation of studies that recognize the potential needs of trauma-exposed participants or communities are described in the Belmont Report [22]. The Belmont Report was released in 1978 and outlines three principles: respect for persons, beneficence, and justice. Respect for persons recognizes the autonomy of potential research participants and limits to the ability of some persons to consent to research participation. Beneficence instructs researchers to not harm participants and to seek to increase positives and limit negatives of research participation. Justice concerns

whether research participants are selected from the populations who are likely to benefit from results. This report was revolutionary when it was released, and the three main principles remain relevant today. However, when the Belmont Report was published, scant research had been conducted on participants' comfort discussing trauma in a research setting, or on what benefits participation in research on trauma may offer participants [23]. Research since the Belmont Report was published indicates that participants can safely discuss trauma in health care [24, 25] and research settings [23, 26–31] without causing undue harm, and that participation in research that addresses trauma may be meaningful and beneficial for trauma-exposed participants [32–34]. In light of our expanded understanding of the pervasiveness and complexity of trauma, we wanted to examine what relevant guidelines and recommendations exist for public health researchers on conducting trauma research and research with trauma-exposed individuals and communities.

This review identifies, describes, and summarizes recommendations for conducting public health research on trauma or with trauma-exposed populations. The key questions are:

1. Are there existing recommendations for conducting research with populations who have experienced trauma? If so, what are they?

2. What gaps exist for public health practitioners to be able to plan and implement trauma-informed research studies?

## Methods

This analysis consisted of a scoping review of four peer-reviewed databases (Embase, PubMed, Scopus, and Web of Science) and two grey literature databases (Open Grey and Google Scholar) from 1978–2020. Scoping reviews are appropriate when the research questions guiding a literature review are not well-defined and when studies of diverse methodologies may be reviewed [35]. We chose the start date of 1978 due to the Belmont Report being released that year, and we wanted to ensure that articles had the ability to incorporate the Belmont principles. We originally conducted searches between November and December 2018 and updated the search in December 2020. We uploaded all results into Endnote and eliminated duplicates. We reviewed titles, abstracts, and full-text using Covidence software for systematic reviews [36], which facilitated double review by our team.

Our review was not accepted into the PROSPERRO systematic review registry because our outcome of interest pertained to research conduct rather than a specific human health outcome. We generated our search string (S1 File) by including synonyms for trauma and ethics, research, or frameworks. We created the search string in collaboration with a health sciences reference librarian and tested it several times to identify the most appropriate set of relevant terms to yield a large number of results. Some of these relevant keywords included "trauma," "trauma-informed," "survivor of trauma," "vulnerable population," "principles," "guidelines," "trauma research design," and "distress protocols." We utilized Boolean operators to exclude physical traumas, such as "traumatic brain injury" and "fracture."

We identified 6,234 references for title/abstract screening by two independent reviewers (** and **) in our 2018 search. A third, independent reviewer (**) resolved conflicts. This resulted in 174 articles for initial (2018) full text review. Two independent reviewers (** and **) reviewed each full text and a third, independent reviewer (**) resolved conflicts.

We used the same search string to update our review in December 2020 using the same databases, limiting our search to articles published in 2019 or 2020. We identified 2,306 references published since our 2018 search and eliminated duplicates, which resulted in 1,835 references for title/abstract screening (all authors). As we conceived our 2020 search would update

our 2018 search with a limited number of articles that may change our framework, a single reviewer screened each title/abstract. This resulted in 126 articles for (2020) full text review. Three reviewers (**, **, and ** reviewed the full texts so that each was independently rated by two reviewers, or a third in cases where two reviewers disagreed.

We included all articles of all study designs in both searches. Exclusion criteria were the same across both searches. Articles were excluded if they were not in English, did not focus on trauma or a trauma-exposed population, and did not include either guidelines or recommendations for researchers about how to conduct research on trauma or with trauma-exposed populations.

From the final set of articles in our 2018 and 2020 searches (S1 File), we extracted information on population, study design or article/report type, type(s) of trauma experienced, setting, and guidelines/recommendations from the authors. Because we were not extracting quantitative data from the articles, we did not assess risk of bias in individual studies or across studies, and we did not use summary measures, as is common in reviews utilizing Preferred Reporting Items for Systematic Reviews and Meta-Analyses guidelines [37]. We analyzed themes from the initial search (in 2018) using MaxQDA [38] to group the text of recommendations into categories based on what part of the research process they applied to, population, and setting. We first read extracted suggestions from ten articles and each co-author drafted potential codes in memos. We then met as a research team, developed a pilot codebook from the group's memos, and coded ten additional extracted suggestions. We collaboratively identified the following as themes from the recommendations and included them in our codebook (available upon request): Institutional Review Board guidance, training, community-informed research, study stages, data collection, researcher characteristics, referral services, protocols, participants, participant empowerment, participant distress, special populations, children, and secondary traumatization.

We resolved differences through discussion and reached consensus on a final codebook. Two separate members of the research team coded the remaining transcripts from the initial 2018 search to arrive at a final set of codes. Following coding, we wrote summaries of each code and used these to produce a narrative summary for the results. We then read across these themes and created tables summarizing recommendations for the following research stages: research design, recruitment, consent, data collection, and results dissemination (available upon request). Because several recommendations cut across research stages and we noticed emergent themes in the set of extracted guidelines, we did a second round of classification sorting the recommendations into the following categories: community benefit, participant benefit, safety, researcher well-being, and the nature and scope of trauma research. Several articles offered recommendations for research with specific populations, and we translated these into broader suggestions where possible.

When we created tables of the recommendations organized by research stage from our 2018 search, each author also produced a list of questions that the recommendations raised for them. One author (**) then identified themes within the set of questions and consolidated these lists, dropping questions that are already widely discussed within the broader research literature (e.g., questions pertaining to community based participatory research (CBPR)). He grouped the retained questions into the following emergent themes on gaps specifically relevant to the conduct of trauma-informed research: research design and conduct, systemic issues, and research translation/applicability.

Because publication themes in our 2020 search were similar to article themes in our 2018 search, we did not code articles in our 2020 search in MaxQDA [38]. We directly sorted them into categories of community benefit, participant benefit, safety, researcher well-being, and the nature and scope of trauma research, as well as research design, recruitment, consent, data

collection, and results dissemination. We did not generate questions in response to recommendations in the articles within our 2020 search as these articles did not offer substantially different recommendations than those identified in our earlier search.

## Results

After removing duplicate articles using the automated processes in EndNote and Covidence and manually removing additional duplicates, we had 8,075 articles (6,234 in 2018 and 1,835 in 2020). After screening titles and abstracts, we had 300 articles for full text review (174 in 2018 and 126 in 2020). Co-authors conducting the full text review excluded 155 articles (96 in 2018 and 59 in 2020) for the following main reasons: Primary purpose of the article not to present guidelines or recommendations (129 articles), Only Clinical Care (19 articles), Book/ Book Chapter (9 pieces), Not Related to Trauma or Traumatized populations (4 articles), Language other than English (3 articles). Thus, we used abstracted data from 145 articles in this review (Fig 1).

The reviewed articles (S1 File) were heterogenous and consisted of a mix of non-research articles (46 articles), qualitative research (37 articles), review pieces (18 articles), mixed methods studies (15 articles), experimental studies (13 articles), cross-sectional studies (5 articles), and observational studies (1 article). The articles studied or commented on the impacts of trauma or trauma research ethics among diverse populations including, children (20 articles), service providers or researchers (19 articles), survivors of other forms of violence or abuse (11 articles), Indigenous people (9 articles), survivors of sexual assault (9 articles), women (9 articles), people with disabilities (7 articles), populations described by sample design (e.g., probability sample) (7 articles), people with unspecified trauma (6 articles), students (6 articles), gender and sexual minorities (4 articles), immigrants and refugees (4 articles), elders (3 articles), incarcerated individuals (2 articles), individuals affected by COVID-19 (2 articles), individuals in 'the Global South' (2 article), New York City residents after 9–11 (2 articles), survivors of intimate partner violence (2 articles), autistic people (1 article), emergency room patients (1 article), individuals who may be both survivors and perpetrators of violence (1 article), individuals affected by natural disasters (1 article), internet users (1 article), people using methamphetamines (1 article), residents in a rural community (1 article), survivors of trafficking (1 article), survivors of war (1 article), and veterans (1 article). The remainder of our Results sections address our findings by each of our two research questions.

### Research Question 1: What are existing recommendations for conducting research with populations who have experienced trauma? (Table 1)

Many of the recommendations we extracted pertain to more than one research stage, and each research stage has multiple recommendations pertaining to each category (S1 File). As a result, we chose to present narrative summaries of the recommendations pertaining to community benefit, participant benefit, safety, researcher well-being, and the nature and scope of trauma research in our Results section. Our full list of reviewed articles (S1 File) also organizes articles by community benefit, participant benefit, safety, researcher well-being, and the nature and scope of trauma research.

**Community benefit.**   Trauma harms not only individuals who experience it, but also the communities that they are members of and/or constitute, and subsequent generations [140, 141]. This review resulted in many recommendations that researchers support community well-being or empowerment throughout the research process, and many articles explicitly addressed systematic inequity [54, 57, 61, 63–65]. Most recommendations endorsed CBPR principles either explicitly or implicitly [142], including considering strengths and protective

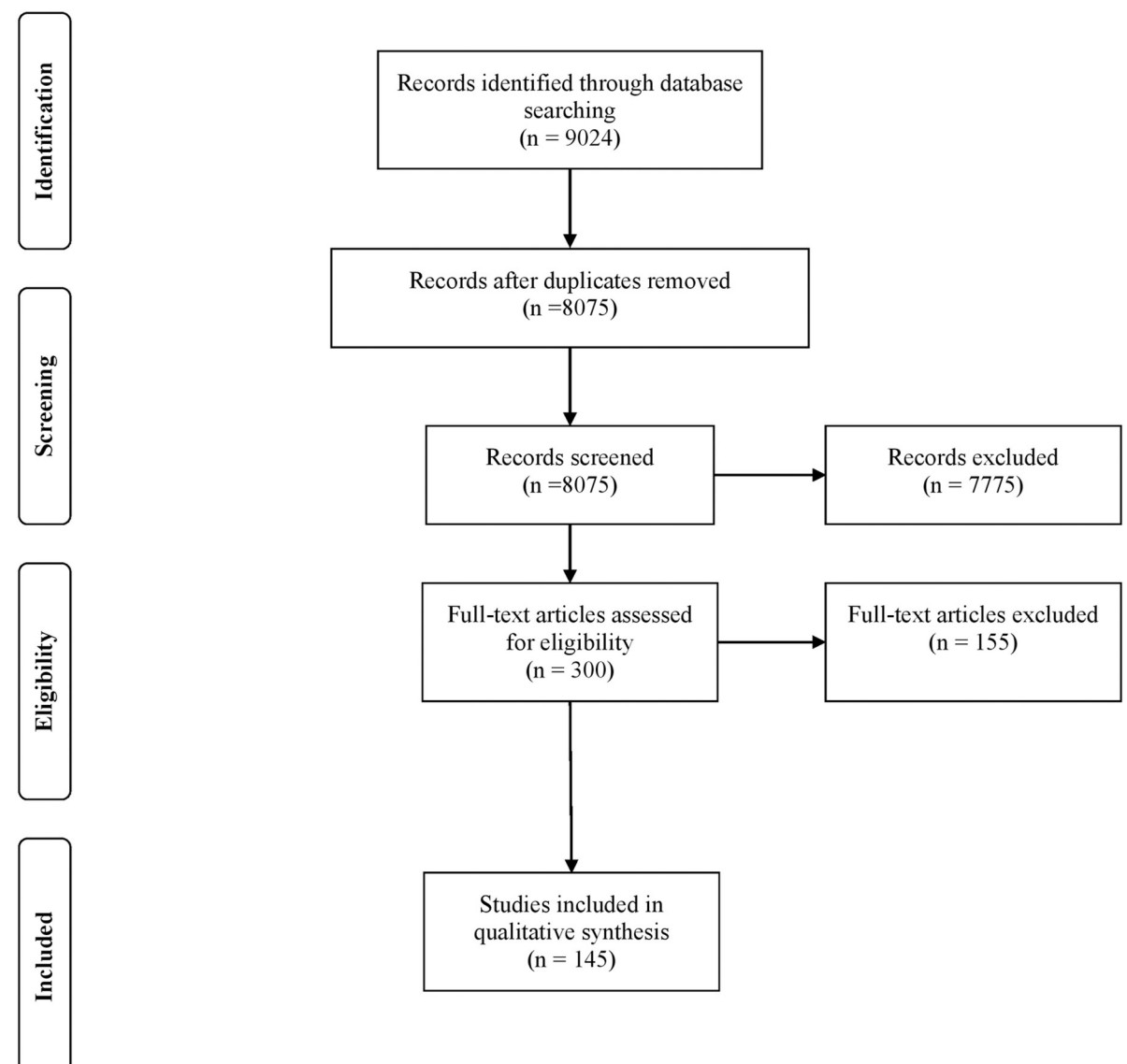

**Fig 1. PRISMA flow diagram for 2018 and 2020 review conducted in the United States on research with trauma-exposed populations.**

factors in community rather than just deficits [52, 56, 57, 61, 63, 64]. Recommendations also supported representation in research (as participants, during study design and ethical review, and during dissemination) of marginalized and trauma-exposed populations in order to ensure research equity [42, 43, 50–52, 57, 62–64, 66]. For thoughtful examples of CBPR in trauma research please see Andrews, Pepler, & Motz (2019) [67], Nnawulezi et al. (2019) [62], and Roche et al. (2020) [63]. Twis & Preble (2020) offer a theoretical framework that may be used with CBPR to address systemic inequity [65]. For a more detailed discussion of community benefit and policy from a public health perspective please see Bowen and Murshid (2016) [41]. For more detailed discussions of historical trauma and community benefit and research when working with or for Indigenous peoples please see Hamby, Elm, & Schulz (2019) [57] (historical trauma), Caldwell et al. (2005) [52], Roche et al. (2020) [63], and Turpel-Lafond & Chondoma (2019) [64].

**Table 1. Select recommendations for research with trauma-exposed populations by research stage, in 2018 and 2020 review conducted in the United States on research with trauma-exposed populations.**

| Category | Recommendation | Research design | Recruitment | Consent | Data collection | Results dissemination |
|---|---|---|---|---|---|---|
| Community Benefit | Focus research on change and use community-based participatory research methods: promote participant agency; do not exclude people with trauma; support empowerment; address stigma, historical trauma, and systematic injustice; involve community partners and participants in oversight of your research from inception to completion—including data interpretation and dissemination, authoring publications, and ownership of the data-; build lasting relationships with communities; create tangible products that benefit the community over the course of your project; embed research findings in historical and cultural context for accurate interpretation; extend confidentiality to the whole community; and consider policy and interventions when you design your research [27, 39–68]. | X | X | X | X | X |
| Participant Benefit | See consent as a dynamic and on-going process—allow consent to be given in multiple formats and seek consent multiple times [27, 30, 43, 44, 51, 69–75]. | X | X | X | | |
| | Consider which method of data collection may be more beneficial for participants (e.g., online, in-person, interview, questionnaire, alternative methods such as photovoice.) When you debrief, talk about the prevalence of the trauma studied and make efforts to de-stigmatize it [74, 76–78] | X | | | X | X |
| | Give participants safe opportunities/spaces to tell stories rather than avoiding disclosure and allow them to tell their story at their own speed. Discussing trauma in a safe environment may be beneficial for participants as it allows self-reflection and meaning-making. Participants may like that their contribution can help others through de-stigmatization, improving resources, and educating others [27, 30, 31, 42, 44, 74, 75, 78–81]. | X | X | X | X | X |
| Safety | Know that safety is socially produced and assess how oppression and cultural, historical, racial, and systemic trauma may affect participants' ability to safely and comfortably engage in research [51, 82–90] | X | X | X | X | X |
| | Train researchers to recognize and respond to unsafe situations for participants. Ask participants for a safe method of contact, when they can be contacted, and if a message can be left, and re-ask because it may change. Assess and support participants in their stages of change. Always assess threats to participant safety (e.g., abuse) and help with safety plans and plans to stop abuse. Get a certificate of confidentiality but be upfront about mandatory reporting. Follow professional standards for documenting abuse and participant referrals. If a participant discloses that they are abusing someone, express an expectation that they will stop it and offer support for them to do so [27, 43, 72, 87, 90–93] | X | X | X | X | X |
| | Assume that your recruitment material will be seen by abusers or people within a participant's social network. Have recruitment material and the study voicemail message be vague enough to not arouse suspicion. When you call or message do not assume that the individual responding is the participant. Always ask if they can communicate freely at that time. Do not leave study-related messages. Do not give more information about the study until you know that it is safe to do so [43, 72, 86, 91, 93] | X | X | X | X | X |
| | Find culturally and politically acceptable support available locally. Offer participants referrals even if they choose not to participate. Have protocols to respond to participant distress and to connect participants with psychological resources. Know staff at agencies for referrals, bring them in to meet the researchers, and quality check referrals provided. Include domestic violence resources in a sheet of other resources and do not place them next to each other so as to not arouse spouse suspicion [27, 30, 31, 43, 51, 71–73, 81, 86, 87, 92, 94–98]. | X | X | X | X | X |
| | Let participants know the goals of the study, why you are doing it, that they don't have to participate, and that topics covered may be distressing. Consider providing examples of potential upsetting questions. Ask if it is safe for a participant to have a copy of the consent form, let them know that their involvement does not affect what services they receive outside of the research study, and inform them as to how their data may be used. Do not make research incentives contingent upon disclosing any information, in particular information that may put undocumented people at risk. Try to provide cash compensation if possible since it is more helpful for low-income people than gift cards [27, 30, 42–44, 50, 51, 69–75, 82, 87]. | X | X | X | X | X |
| | Check in with participants throughout data collection and let them set a pace that works for them. Be ready to stop data collection if necessary and have a backup plan. If not collecting data in person, consider having a code word or phrase that a participant can use to indicate if they are unsafe and have to stop. Consider collecting data over multiple sessions so participants can modulate their exposure. Make sure you allow for time between sessions for participants to recoup and explain how subsequent data collection sessions may be different. Normalize and de-stigmatize conversations about and reactions to trauma. Create multiple opportunities for disclosure but minimize the need for a survivor to repeat their stories multiple times. Be prepared to answer participant questions throughout your engagement with them and be ready to be open share but maintain boundaries. Assist participants with closure if necessary [30, 31, 41, 43, 49, 50, 71–75, 78, 79, 81, 82, 86, 90, 97–101]. | X | X | X | X | X |
| | Do a debriefing even if distress is not evident. If using forensic-style interviews, ensure that participants understood the questions (especially children). If interviewing a child, debrief afterwards to help them differentiate the real and experimental situations that the interview discussed [44, 73, 97]. | X | | | X | X |
| Researcher Well-Being | Research staff should be supervised by individuals who are trauma-sensitive and up-to-date with trauma literature and care, who can model appropriate boundaries and coping [42, 74, 102–108]. | X | X | X | X | X |
| | Train research staff on countertransference and how to discuss trauma with participants. Also train them on how the study may affect them (consider both secondary trauma and researcher trauma history), differential risks of secondary traumatization, de-briefing, self-care and methods to recognize their own distress such as journaling. Keep a reflective journal [92, 102–114]. | X | X | X | | X |
| | Acknowledge the important role that researchers with a trauma background play in trauma research, what benefits conducting trauma research may have for them, and what supports they may need [31, 104]. | X | X | | X | X |
| | Make sure researchers pace their workloads; break up interviews into multiple sessions if necessary; account for the possibility of participants sharing something that challenges how you see the world (or understand your own trauma if you have a trauma history) and plan time for self-care, procrastination and reflection [31, 97, 102, 103, 105, 107, 111]. | X | X | | X | X |
| | Researchers need to talk about emotion in the research process and support each other [105, 106, 113]. | X | | | X | X |
| | IRBs need to consider not just participant risk but researcher risk as well [103, 106, 111, 114]. | X | | | | X |
| | Provide access to crisis counseling or other venues for researchers to process their experience safely [92, 103–105, 108, 112]. | X | X | | X | X |
| | Disseminate research as a way of processing its impact; make sure you have support to use your data and make your results accessible and available to multiple audiences. Potentially use for political action or policy change as appropriate [103]. | X | | | | X |

*(Continued)*

**Table 1.** (Continued)

| Category | Recommendation | Research design | Recruitment | Consent | Data collection | Results dissemination |
|---|---|---|---|---|---|---|
| **Nature and Scope of Trauma Research** | Consider trauma symptoms of shame and dissociation as meaningful predictors and intervention points. [31] Know that research is not inherently harmful and may be beneficial. Distress may indicate engagement and does not necessarily mean harm. Consider distress in terms of minimal risk (e.g., does it exceed what is experienced in life outside research) and consider whether distress is an increase in existing symptoms or if it is a new emotion for participants. Consider the possibility of a nocebo effect and the possibility that this could encourage trauma avoidance. Collect information about the experience of participating in research, using validated tools such as the Reactions or Research Participation Questionnaire [115]. Consider benefits of participation in research, and examine the long-term effects of research participation. Consider who should collect this data and how it should be collected to minimize social desirability bias. These actions should not be limited to trauma research since distress is felt in different types of studies and considering more topics, populations, and research designs can increase generalizability. Publish this research so others can learn from it, and educate IRBs, and include information on participant experiences in all IRB applications. All results should discuss participant experiences, benefits and costs of participation, consent, confidentiality, research problems, and effectiveness of solutions tried. Also, IRBs should require that investigators collect data on research experiences, consider benefits of participation, and do systematic reviews on how methods affect distress. Professional organization that work on trauma and research should craft guidelines for how to discuss risk [23, 26, 27, 30, 31, 69, 74, 75, 77–81, 96, 115–120]. | X | | X | X | X |
| | Research participant-researcher relationships as well as participant-participant experiences. Examine factors that predict or explain why some participants experience distress and others do not [23, 29, 31, 75, 77, 80, 81, 96, 118]. | X | | X | X | X |
| | When advertising your study account for the possibility that some people may not remember or identify their trauma as abuse; develop protocols for trying to prevent such participants from ending up in your control group so as to not affect your results (e.g., ask about specific experiences and screen for trauma symptoms). Know that retrospective and prospective measures of trauma may measure different constructs and consider that people with trauma may differ from people without trauma in terms of pre-existing vulnerabilities. Also consider that some participants may be survivors and perpetrators of trauma. When screening participants do not ask them to self-identify or label their experiences, instead ask they about specific experiences they may have had and be aware of any mandating reporting standards [31, 93, 121–123]. | X | X | | | |
| | Consider disclosure and trauma symptoms of shame and dissociation as meaningful predictors and intervention points. Also investigate the utility of simpler versus more comprehensive interventions, research mediators and moderators, and invest in developing and using standardized measures [31, 124–128]. | X | | | X | X |
| | Examine factors that predict or explain why some participants experience distress and others do not [23, 29, 31, 75, 77, 80, 81, 96, 118]. | | | | | |
| | Seek community and public involvement in research and investigate practical and ethical challenges in this research. Researchers who belong to populations that experience marginalization may seek to affect change through their research, or by providing bridging social capital between researchers and wider communities. Bonding social capital is required to do this. Researchers who are complete outsiders to a community should seek to partner with researchers who are also community insiders. Intersectionality [129], reflexivity, and community based participatory research are helpful tools [130–139]. | X | | | X | X |

*An expanded version of this table with all recommendations in the review is available upon request.

**Participant benefit.** Trauma is a fundamental experience of disempowerment [143, 144]; thus, researchers who seek to redress trauma must attend to the disempowerment that survivors experience. Participant benefit recommendations illustrated ways that researchers can support participant well-being or empowerment in the process of conducting research. These recommendations highlight methods for making participants feel more comfortable throughout a study [53], suggestions to honor participant autonomy by improving processes of informed consent [27, 30, 51, 73, 92], and consideration of what makes research participation beneficial for participants [23, 27, 31, 44, 74, 77–79, 81]. Recommendations for improving informed consent included suggestions to be transparent about potential risks and benefits [27, 73, 74, 92], and ensuring participant comfort with and comprehension of research consent through means such as seeking consent throughout the research process [51]. Articles discussing factors that make participation in trauma research meaningful for participants highlighted the importance of participants having a venue in which to tell their story safely (for a theoretical treatment see Draucker, Martsolf, & Poole, 2009) [27, 31, 74, 75, 78, 81, 90, 121], and the desire that participants may have to help improve things for others [23, 44, 78, 80, 121]. Additionally, several studies emphasized that participants must have access to -and thus some ownership over- any study results [73, 78, 145].

**Safety.** Trauma fundamentally threatens the physical and/or psychological safety of those who experience it [143]. Many potential participants in trauma research may experience ongoing threats to their safety, and trauma researchers are often concerned about the possibility that discussing trauma may cause participants distress. Safety recommendations addressed ways to assess and minimize the possibility of harm due to research participation that could make participants more vulnerable to violence, or data collection that may cause distress. Articles also considered participant safety as a function of systemic exposures and social identity (e.g., incarceration or immigration status creating pressure to participate, or sexual orientation rendering parental consent for minors to participate in research dangerous) [82–84, 88, 89, 146] Goodwin & Tiderington (2020) present thoughtful trauma-informed recommendations on how to conduct research that take into account participants' social identities and systemic exposures [87]. Mwambari (2019) provides a particularly thick and rich description of the risks that participants and local researchers can incur from working with non-local researchers, as well as recommendations to lessen these risks [84]. For a more detailed discussion on reducing participants' vulnerability to violence or distress from research participation please see: Ahlin (2019) [89], Kyegombe et al. (2019) [146], and Pickles (2020) [83] for recommendations on obtaining assent from minors in vulnerable positions, Ahlin (2019) [89] and Dehghan & Wilson (2018) [88] for recommendations on handling consent or asset when potential participants may feel pressured to participate, Kimberg (2018) [91] and Sullivan & Cain (2004) [43] for recommendations pertaining to domestic violence, Allden et al. (2009) [51] for recommendations pertaining to political violence and stigmatization threats, Amin and Garcia-Moreno (2018) [147] for recommendations pertaining to disclosure of sexual abuse, and Linabary & Corple (2019) [85] for recommendations concerning online research and privacy. Recommendations for minimizing the possibility of harm from data collection distress included providing resources for participants through accountable, acceptable, and available quality referrals (either through lists or proactive referrals) [27, 31, 51, 72, 73, 86, 92, 94–96], and monitoring participants' emotional well-being during and following data collection [30, 44, 71, 73, 75, 79, 82, 97].

**Researcher well-being.** When studying trauma, researchers may experience participants' trauma vicariously, which could cause distress or burn-out [31, 104]. Several authors offered recommendations to minimize distress and promote researcher well-being through self-care and addressing secondary traumatization. These recommendations included pacing one's

workload [103, 108], cultivating one's support network [31, 97, 103, 105, 107], connecting with other trauma researchers [105, 113, 148, 149], and preventing, recognizing, and responding to emotional distress experienced by researchers as a result of conducting research [31, 42, 72, 92, 102–107, 111] Some authors considered researcher well-being not just the responsibility of the researcher, but the responsibility of universities and Institutional Review Boards (IRBs) as well [105, 106, 111, 112, 114]. Recommendations to minimize distress and promote researcher well-being through environmental supports included trauma-informed supervision, de-briefing, and access to counseling [72, 102–108, 111, 112]. For more detailed discussions of secondary traumatization, self-care, and considerations for researchers with a trauma background themselves, please see Eadesa et al. (2020), [104] Fohring (2020 [105], Jeftic (2020) [113], Markowitz (2019) [106], and Wager (2011) [31]. Researchers interviewing perpetrators may want to read Jeftic (2020) [113] and Markowitz (2019) [106].

**Nature and scope of trauma research.**   Articles on the scope and nature of trauma research concerned particular trauma topics, risk/benefits to participation, methodology, and application of theory to trauma research. Particular trauma topics included interventions [125], disclosure [124], cognitive distortions [150], and interviewing children [151]. A number of articles noted that participation in trauma research often posed a low risk of harm to participants [26–29] and may even be beneficial to individuals who have experienced trauma [23, 28, 30, 80]. Some articles also pointed out that distress during participation does not necessarily indicate that harm has been done [23, 30, 31]. Readers may wish to refer to Newman & Kaloupek (2004) for a discussion on how trauma research does not re-traumatize participants [23]. Nonetheless, several authors recommended increasing research on the effects of participation in trauma research [23, 26, 30, 31, 74, 75, 77, 80, 81, 118, 152] as well as educating IRBs on the possible benefits of trauma-focused research [23, 77, 80]. Articles on methodology concerned sampling and data collection [153–156], a need to use consistent measures and not conflate different constructs [123, 127, 128], a need to see participants as potentially being both survivors and perpetrators of violence [122], and the nocebo effect in trauma research [119]. Researchers offered recommendations on improving internal research validity when potential control group members with trauma do not identify themselves as having trauma [31, 121], and many recommended using participatory research methods to conduct better research [132, 135, 137, 138, 155]. Gultekin et al. (2019) [133] introduced the eco-social trauma intervention model, which may be used across disciplines to develop and pilot trauma-informed interventions, and Tol (2020) [139] described using a social justice framework for research. A few articles concerned research and insider-outsider dynamics [130, 131, 136]. For example, Carr (2019) [130] describes situated solidarity as a possible approach for academics who also belong to communities that the focus of their research proposes to benefit, and Gaillard & Peek (2019) [131] recommend a structure for outsider and local researchers to work together in the aftermath of disaster.

## Research Question 2: What gaps exist for public health practitioners to be able to plan and implement trauma-informed research studies?

Following our review of existing recommendations, we identified a select number of gaps specific to the literature on trauma research that, should they be addressed, may advance public health research on trauma or with trauma-exposed populations. The gaps we identify are far from exhaustive, but we hope they may catalyze further conversation and future scholarship. We acknowledge that work is being done to address the gaps we identify, and we hope to see this work continue. We comment briefly on three themes pertaining to gaps below (research

design and conduct, systemic issues, and research translation/applicability) and provide more detailed explanation and a select number of questions to consider in (S1 File).

**Research design and conduct.**    We identified a need for additional scholarship on: 1) how to apply theory in the design of trauma-relevant research; and 2) how research populations should be defined and investigated in trauma research. The first is a need because effective use of theory in research facilitates knowledge building and communication between researchers [157], and can also inform intervention design and testing [158]. Without additional guidance on how to apply theory in trauma-relevant research, scholars may utilize theories in a fragmented manner that does not support knowledge unification. The second is a need because insufficient consensus on who is traumatized and how to define their exposure limits both epidemiological and intervention research. For example, people of color and other marginalized groups are under-represented in psychological trauma intervention research [159–161] which limits the generalizability and applicability of psychological trauma treatment research [159, 160, 162].

**Systemic issues.**    We identified a need for research on psychological trauma that: 1) utilizes multi-level frameworks [163, 164]; 2) conceives and attends to multiple dimensions of trauma simultaneously (e.g., historical, cultural, racial, and systemic trauma, plus individually-oriented conceptions of trauma); and 3) focuses on the role that systemic injustices (e.g., institutionalized racism and colonialism) and abuse of institutional power (e.g., state-sponsored torture or terror) have in shaping trauma exposures and responses. These are gaps because trauma exposure, immediate harm done by trauma, and societal responses to trauma are largely determined within complex social systems [17]; as such, research that does not account for multiple social ecological exposures, multiple dimensions of trauma, or societal power dynamics, may be ineffective at addressing trauma epidemiology and interventions. They may even promote distorted perceptions of and inappropriate responses to trauma.

**Research translation/applicability.**    We identified two pressing needs for scholarship pertaining to research translation or applicability: 1) literature should address how to translate findings to wider -and multi-ethnic-[160] audiences and how to scale up evidence-based or promising interventions; and 2) researchers need guidance on how to design or use research to effectively impact policy. The first is a need because the lag between trauma-relevant research and public knowledge regarding trauma creates treatment barriers for trauma survivors and promotes public disinvestment in trauma prevention and treatment research [165, 166]. The second is a need because policy is an important determinant of health [167] (that many sectors are presently seeking to leverage to prevent and treat trauma), but health researchers are not typically trained to design studies with immediate relevance to informing policy or stakeholders working within cross-sectorial partnerships to influence policy.

## Discussion

Although a robust body of public health research on trauma and with trauma-exposed people and communities exists, gold standards for the conduct of trauma-informed public health research have not yet been authored. Many researchers draw upon principles described in the Belmont Report to inform the ethical conduct of their research. While the Belmont Report offers guiding principles for the ethical conduct of research that remain relevant today, its principles are necessarily part of a larger living standard of research ethics that should be evolving over time. Since the Belmont Report was published, the Council for International Organizations of Medical Sciences published "International Guidelines For Ethical Review of Epidemiological Studies" [168] in 1991, and updated "International Ethical Guidelines for Epidemiological Studies" [169] in 2009. The World Health Organization (WHO) has also

published "Putting Women First: Ethical and Safety Recommendations for Research on Domestic Violence Against Women" [170] in 2001, "WHO Ethical and Safety Recommendations for Researching, Documenting and Monitoring Sexual Violence in Emergencies" [171] in 2007, and "Ethical and Safety Recommendations for Intervention Research on Violence against Women" [172] in 2016. While these guidelines do elaborate on the ethical conduct of research with vulnerable populations, and do consider the importance of access to research participation for diverse groups, gaps remain in the ethics of inclusion in research [160, 173] and they do not provide guidance on the conduct of trauma-informed research that may be applied across populations [169]. CBPR has also emerged as an ethical approach to research since the publication of the Belmont report, and offers an extensive set of guidelines for working with communities that may inform research with trauma-exposed populations or on systemic, cultural, racial, or historical trauma. However, studies may be able to be trauma-informed without implementing full CBPR standards, which can be resource-intensive and time-consuming.

Despite the current lack of gold standards for trauma-informed research, this review identified recommendations in the literature for trauma research or research with trauma-exposed populations. These recommendations include using research for change (e.g., advocacy and de-stigmatization) [27, 31, 39–45, 49, 50, 56, 57, 60, 62, 64, 66, 68, 78, 84, 86–88, 90, 91, 95, 98, 101, 103, 135, 138, 174], centering agency and growth (e.g., assess and adapt to participants' needs for agency and safety in an ongoing manner, and provide appropriate support and resources) [27, 30, 31, 42–44, 51, 55, 61, 63, 67, 71–81, 90, 91, 93–95, 99–101, 136, 175], preparing and supporting researchers to process the impact of trauma connected to research [31, 92, 97, 102–113, 176], and improving professional standards (e.g., study short and long-term effects of research participation, identify factors influencing research experiences, minimize the inclusion of trauma-affected control participants, and acknowledge meaningful predictors and intervention points) [23, 26, 27, 29–31, 65, 69, 74, 75, 77–81, 85, 93, 96, 115–119, 121–123, 125, 127, 128, 131, 133, 139, 146, 151, 154, 177]. We also identified needs for guidance and continued discussion pertaining to the design and conduct of research (e.g., how to use theory and how to define and investigate research populations), systemic issues (e.g., how to use multi-level frameworks and multi-dimensional understandings of trauma, and how to address systemic injustice), and research translation and applicability (e.g., how to translate findings widely and scale up evidence-based interventions, and how to impact policy.).

This review, while contributing to literature, still has several limitations. First, though we used current literature and worked closely with a health sciences research librarian to define our search string, there could be important terms we missed. For example, we did not include "complex trauma" or "developmental trauma" in our search string, and we did not search the PsychINFO database. In addition, due to the large number of references reviewed, we did not add articles to our review from the reference lists of the articles we found through our search string. Additionally, due to our restriction to English-language articles, many articles we reviewed presented research based in the United States, and further research on trauma globally is necessary. Finally, we completed a thorough qualitative analysis on the final set of articles, but additional coders or a more extensive reconciliation process mays have further refined our findings.

This review also incorporates several strengths. First, by using a systematic, replicable search strategy, future researchers can recreate or easily expand on our methodology. Second, we attempted to incorporate non-peer reviewed and grey literature sources by including two grey literature databases in our search. Third, our research team draws on a diverse set of content and professional expertise (immigrant health, substance use, historic and systemic trauma,

intimate partner violence, medicine, nursing), allowing for incorporation of distinct perspectives into the search string, strategy, and qualitative analysis.

It is not the goal of this review to create guidelines; however, developing specific and universal trauma-informed research guidelines that could be implemented and tested would improve research quality and support the mentoring of early career researchers. Researchers and community partners could use gold standards that provide a customizable "menu" of guidelines to inform protocol development, data analysis, dissemination, and application of results. This article highlights the need for such guidelines to be authored. Given public health's history of unethical research (e.g. Tuskegee Syphilis Trials) [178] that prompted publication of the Belmont Report, combined with the potential public health research has to be a vehicle of community and individual empowerment, we contend that individuals and communities affected by trauma must be involved in authoring a set of gold standard guidelines. We also contend that such guidelines must provide actionable and evaluated steps to increase the participation of people of color and other marginalized populations in research, in order to be relevant for and accountable to a majority of trauma-exposed populations [160]. We comment that gaps pertaining to research design and conduct, systemic issues, and research translation/applicability require additional inquiry. These gaps are not exhaustive and scholarship is addressing them. Nonetheless, additional conversations about gaps and how to address them undoubtedly are needed, and those conversations must involve both researchers across disciplines and those affected by trauma. Institutes that are critical to the conduct of public health research, such as the National Institutes for Health, could gather individuals and communities affected by trauma to address gaps in a more comprehensive manner, and to author guidelines for the conduct of trauma-informed public health research. Given that public health researchers are increasingly embracing and researching trauma as a determinant of health, such guidelines and focus on gaps would be appropriate and timely.

## Supporting information

**S1 File.**
(DOCX)

**S1 Data.**
(XLSX)

**S1 Checklist.**
(DOC)

## Acknowledgments

We wish to thank the following individuals for reviewing and offering input on this article, including our identification of recommendations and gaps (listed in alphabetical order): Christa Krüger, Michelle Sotero, Monica Williams. We also thank Shenita R. Peterson, who assisted us in developing an initial search string and search strategy. We also acknowledge receipt of professional development funds from Emory University's Laney Graduate School to Kaitlyn Stanhope, which was used for purchase of systematic review software.

## Author Contributions

**Conceptualization:** Kevin Jefferson, Kaitlyn K. Stanhope, Carla Jones-Harrell, Aimée Vester, Emma Tyano, Casey D. Xavier Hall.

**Data curation:** Kevin Jefferson, Kaitlyn K. Stanhope, Carla Jones-Harrell, Aimée Vester, Emma Tyano, Casey D. Xavier Hall.

**Formal analysis:** Kevin Jefferson, Kaitlyn K. Stanhope, Carla Jones-Harrell, Aimée Vester, Emma Tyano, Casey D. Xavier Hall.

**Funding acquisition:** Kaitlyn K. Stanhope.

**Investigation:** Kevin Jefferson, Kaitlyn K. Stanhope, Carla Jones-Harrell, Aimée Vester, Emma Tyano, Casey D. Xavier Hall.

**Methodology:** Kevin Jefferson, Kaitlyn K. Stanhope, Carla Jones-Harrell, Aimée Vester, Emma Tyano, Casey D. Xavier Hall.

**Visualization:** Kevin Jefferson, Kaitlyn K. Stanhope, Carla Jones-Harrell.

**Writing – original draft:** Kevin Jefferson, Kaitlyn K. Stanhope, Emma Tyano.

**Writing – review & editing:** Carla Jones-Harrell, Aimée Vester, Casey D. Xavier Hall.

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
