## [Decision Letter · Decision Letter 0]

9 Dec 2020

PONE-D-20-21597

A Scoping Review of Recommendations in the English Language on Conducting Research with Trauma-Exposed Populations Since Publication of the Belmont Report

PLOS ONE

Dear Dr. Jefferson,

Thank you for submitting your manuscript to PLOS ONE. After careful consideration, we feel that it has merit but does not fully meet PLOS ONE’s publication criteria as it currently stands. Therefore, we invite you to submit a revised version of the manuscript that addresses the points raised during the review process.

Thank you for your submission on the very important topic of best practices for conducting research with trauma-exposed populations. Unfortunately, throughout the COVID-19 pandemic it has been difficult securing reviewers. The reviewer secured was very favorable about your manuscript. I have included some minor revisions below which could help improve the quality of the manuscript before publication.

We look forward to receiving your revised manuscript.

Kind regards,

Michelle L. Munro-Kramer, PhD, CNM, FNP-BC

Academic Editor

PLOS ONE

Journal Requirements:

2. Please consider updating this review to allow the inclusion of studies published within the past 12 months.

3.We note that you have indicated that data from this study are available upon request. PLOS only allows data to be available upon request if there are legal or ethical restrictions on sharing data publicly. For information on unacceptable data access restrictions, please see http://journals.plos.org/plosone/s/data-availability#loc-unacceptable-data-access-restrictions.

5.Thank you for stating the following in the Financial Disclosure section:

[KS, No grant number, Laney Graduate School Professional Development Support Funds, https://www.gs.emory.edu/professional-development/pds/index.html, No].   

We note that one or more of the authors are employed by a commercial company: Independent Researcher, Atlanta, Georgia, United States

Additional Editor Comments (if provided):

1) Please ensure that you are using page numbers with direct quotes. There are a number of direct quotes in the introduction.

2) The PRISMA diagram (and text on page 8) notes that the full text of 174 articles were reviewed and 96 were excluded for various reasons. This equates to a total of 78 (not the 69 listed in the text and PRISMA diagram). Please explain.

2) Please consider using a scoping review methodology to frame the methods section (e.g., Arksey & O'Malley, 2005) and describe why a scoping review was appropriate for this topic.

3) Please introduce the use of Covidence in the methods section. It only appears once in the first line of the results.

4) Table 1 is very dense and it is difficult to distill down the main points from the recommendations section. Could this be distilled down to succinct major recommendations that could be further described in the text? The information in this table is extremely useful, but in its current form I am worried it will be easily overlooked because of the density of the text.

5) There are a number of World Health Organization documents on ethical research related to violence and trauma: (a) Putting women first: Ethical and safety recommendations for research on

domestic violence against women (2001); (b) WHO Ethical and safety recommendations for researching, documenting and monitoring sexual violence in emergencies (2007), (c) Ethical and safety recommendations for intervention research on violence against women (2016). I recognize that this manuscript takes a broader approach to trauma and that these documents may not have appeared in the search. However, I think it is worth noting that these international guidelines exist.

6) Most of the articles included in this review were US-based research. This should be addressed in the discussion about future implications for consideration of trauma research within global settings.

Reviewers' comments:

Reviewer's Responses to Questions

**Comments to the Author**

1. Is the manuscript technically sound, and do the data support the conclusions?

Reviewer #1: Yes

2. Has the statistical analysis been performed appropriately and rigorously? 

Reviewer #1: N/A

3. Have the authors made all data underlying the findings in their manuscript fully available?

Reviewer #1: No

4. Is the manuscript presented in an intelligible fashion and written in standard English?

Reviewer #1: Yes

5. Review Comments to the Author

Reviewer #1: This is a well conceived paper that is much needed by researchers and members of ethics review boards. The authors have taken the first large step in bringing together the vast range of ethical issues inherent in researching sensitive topics or when researching potentially vulnerable groups. The categorization of the different issues/solutions is a useful conceptual tool to add the development of socially justice research protocols. There is a exceptional transparency and rigor evident in the methodology and a concise and coherent results section despite the inclusion of a fairly large number of papers in the review. As a reviewer it is rare to see a paper that you think is ready for publication, yet this is how I feel about this particular manuscript. It is well written and deserves widespread dissemination.

6. PLOS authors have the option to publish the peer review history of their article (what does this mean?). If published, this will include your full peer review and any attached files.

Reviewer #1: **Yes: **Nadia Wager

---

## [Author Response · Author response to Decision Letter 0]

9 Jun 2021

Dear Editors,

Thank you kindly for reviewing our scoping review on the conduct of trauma research and offering us an opportunity to improve our manuscript. Per suggestion, we have updated our review through to 2020. We will upload the minimal anonymized data set necessary to replicate study findings as a Supporting Information file. Below we have addressed feedback from our reviewer:

Additional Editor Comments (if provided):

1) Please ensure that you are using page numbers with direct quotes. There are a number of direct quotes in the introduction.

Thank you for pointing this out. We have added page numbers to direct quotes. 

2) The PRISMA diagram (and text on page 8) notes that the full text of 174 articles were reviewed and 96 were excluded for various reasons. This equates to a total of 78 (not the 69 listed in the text and PRISMA diagram). Please explain.

This was an error in reporting on our part. Since we have updated our review, we now have 300 full texts that we have reviewed, 145 of which we have included in our review. We have updated our materials to be consistent now. 

2) Please consider using a scoping review methodology to frame the methods section (e.g., Arksey & O'Malley, 2005) and describe why a scoping review was appropriate for this topic.

Thank you for this suggestion. We added a sentence describing when a scoping review would be used and provided a reference to Arksey & O’Malley, 2005. We feel this is sufficient given that we describe our methodological steps in detail. We are happy to expand our coverage of the scoping review framework though.

3) Please introduce the use of Covidence in the methods section. It only appears once in the first line of the results.

Thank you, we have added this.

4) Table 1 is very dense and it is difficult to distill down the main points from the recommendations section. Could this be distilled down to succinct major recommendations that could be further described in the text? The information in this table is extremely useful, but in its current form I am worried it will be easily overlooked because of the density of the text.

Thank you for this feedback. It is extremely helpful to know how our materials ‘land’ with readers. We have tried to consolidate information in table 1 to make it easier to read and use.

5) There are a number of World Health Organization documents on ethical research related to violence and trauma: (a) Putting women first: Ethical and safety recommendations for research on

domestic violence against women (2001); (b) WHO Ethical and safety recommendations for researching, documenting and monitoring sexual violence in emergencies (2007), (c) Ethical and safety recommendations for intervention research on violence against women (2016). I recognize that this manuscript takes a broader approach to trauma and that these documents may not have appeared in the search. However, I think it is worth noting that these international guidelines exist.

Thanks for pointing these out to us. We have added mention of and reference to them.

6) Most of the articles included in this review were US-based research. This should be addressed in the discussion about future implications for consideration of trauma research within global settings.

This is a limitation. We have added a sentence mentioning this in our discussion section.

---

## [Editor Report · Decision Letter 1]

18 Jun 2021

A Scoping Review of Recommendations in the English Language on Conducting Research with Trauma-Exposed Populations Since Publication of the Belmont Report; Thematic Review of Existing Recommendations on Research with Trauma-Exposed Populations

PONE-D-20-21597R1

Dear Dr. Jefferson,

We’re pleased to inform you that your manuscript has been judged scientifically suitable for publication and will be formally accepted for publication once it meets all outstanding technical requirements.

Kind regards,

Michelle L. Munro-Kramer, PhD, CNM, FNP-BC

Academic Editor

PLOS ONE

Additional Editor Comments (optional):

Thank you for your careful attention to all comments. I look forward to seeing this in publication.
---

## [Editor Report · Acceptance letter]

19 Jul 2021

PONE-D-20-21597R1 

A Scoping Review of Recommendations in the English Language on Conducting Research with Trauma-Exposed Populations Since Publication of the Belmont Report; Thematic Review of Existing Recommendations on Research with Trauma-Exposed Populations 

Dear Dr. Jefferson:

I'm pleased to inform you that your manuscript has been deemed suitable for publication in PLOS ONE. Congratulations! Your manuscript is now with our production department. 

Kind regards, 

on behalf of

Dr. Michelle L. Munro-Kramer 

Academic Editor

PLOS ONE